# Investigating the Modes of Action of the Antimicrobial Chalcones BC1 and T9A

**DOI:** 10.3390/molecules25204596

**Published:** 2020-10-09

**Authors:** Luana G. Morão, André S. G. Lorenzoni, Parichita Chakraborty, Gabriela M. Ayusso, Lucia B. Cavalca, Mariana B. Santos, Beatriz C. Marques, Guilherme Dilarri, Caio Zamuner, Luis O. Regasini, Henrique Ferreira, Dirk-Jan Scheffers

**Affiliations:** 1Departamento de Bioquímica e Microbiologia, Instituto de Biociências, Universidade Estadual Paulista, 130506-900 SP Rio Claro, Brazil; luanagm_bio@yahoo.com.br (L.G.M.); gui_dila@hotmail.com (G.D.); caiozamuner@gmail.com (C.Z.); 2Department of Molecular Microbiology, Groningen Biomolecular Sciences and Biotechnology Institute, University of Groningen, 9747 AG Groningen, The Netherlands; andreglock@gmail.com (A.S.G.L.); p.chakraborty@rug.nl (P.C.); l.bonci.cavalca@rug.nl (L.B.C.); 3Departamento de Química e Ciências Ambientais, Instituto de Biociências, Letras e Ciências Exatas, Universidade Estadual Paulista, 15054-000 SP São José do Rio Preto, Brazil; gabimayusso@gmail.com (G.M.A.); mariana19bsantos@gmail.com (M.B.S.); bbiacarvalho@outlook.com (B.C.M.); luis.regasini@unesp.br (L.O.R.)

**Keywords:** chalcone, Asiatic citrus canker, *Bacillus subtilis*, *Xanthomonas citri*

## Abstract

*Xanthomonas citri* subsp. *citri* (*X. citri*) is an important phytopathogen and causes Asiatic Citrus Canker (ACC). To control ACC, copper sprays are commonly used. As copper is an environmentally damaging heavy metal, new antimicrobials are needed to combat citrus canker. Here, we explored the antimicrobial activity of chalcones, specifically the methoxychalcone BC1 and the hydroxychalcone T9A, against *X. citri* and the model organism *Bacillus subtilis*. BC1 and T9A prevented growth of *X. citri* and *B. subtilis* in concentrations varying from 20 µg/mL to 40 µg/mL. BC1 and T9A decreased incorporation of radiolabeled precursors of DNA, RNA, protein, and peptidoglycan in *X. citri* and *B. subtilis*. Both compounds mildly affected respiratory activity in *X. citri*, but T9A strongly decreased respiratory activity in *B. subtilis*. In line with that finding, intracellular ATP decreased strongly in *B. subtilis* upon T9A treatment, whereas BC1 increased intracellular ATP. In *X. citri*, both compounds resulted in a decrease in intracellular ATP. Cell division seems not to be affected in *X. citri*, and, although in *B. subtilis* the formation of FtsZ-rings is affected, a FtsZ GTPase activity assay suggests that this is an indirect effect. The chalcones studied here represent a sustainable alternative to copper for the control of ACC, and further studies are ongoing to elucidate their precise modes of action.

## 1. Introduction

Brazil is the world leader in sweet orange production, exporting more than 80% of concentrated orange juice consumed in the world. *Xanthomonas citri* subsp. *citri*, the causative agent of Asiatic Citrus Canker (ACC), is a bacterial phytopathogen that can infect all the commercially important species of sweet orange, and is a major threat to the orange juice industry [1]. The most effective way to control the disease is to eradicate infected trees. However, in the current Brazilian legislation, eradication has been substituted by a set of combined measures known as risk mitigation systems, which include periodic sprays of copper compounds to prevent the spread of *X. citri* in the orchards [2]. Spraying trees with copper-containing bactericides has led to the emergence of resistant strains [3]. In addition, copper is stable and accumulates in the soil, which has a negative environmental impact [4].

New antimicrobials are needed to tackle the problem of phytopathogens infecting citrus plants. In this study, we focus on the antimicrobial action of chalcones. Chalcones are plant-derived compounds, consisting of two aromatic rings linked by an α, β-unsaturated ketone (enone bridge), and are precursors of flavonoids and isoflavonoids that belong to plant defence mechanisms. Chalcones prevent damage by microorganisms, insects, and animals, and counteract reactive oxygen species, preventing molecular damage [5]. Many chalcones have been studied for their antimicrobial activity [5,6]. Licochalcone A, naturally produced by the roots of *Glycyrrhiza inflata* has activity against *Bacillus subtilis* and *Staphylococcus aureus*, and is able to prevent their growth at 3 μg/mL. However, this compound shows no activity against gram-negative bacteria [7,8]. Licochalcone A was shown to inhibit oxidation of NADH and oxygen consumption in *Micrococcus luteus* [8], and induce membrane permeability in *S. aureus* [6]. Despite being inactive against gram-negative bacteria, Licochalcone A is able to inhibit NADH oxidases in membranes isolated from *Escherichia coli* and *Pseudomonas aeruginosa*, suggesting that the compound cannot penetrate the outer membrane [7]. Accordingly, some chemically modified chalcones are effective against gram-negative bacteria [5,6], and *Mycobacterium tuberculosis* [9]. The antitubercular activity is associated with inhibition of the protein tyrosine phosphatase B [9]. Phloretin, a chalcone present in apple and kumquat, is effective only against gram-positive bacteria [10]. This compound inhibits acetate dehydrogenase, isocitrate dehydrogenase, and catalase from *S. aureus* [10]. DNA gyrase was reported to be a target for synthetic chalcones active against both gram-positive and gram-negative bacteria [11]. Finally, chalcones have antioxidant, anti-inflammatory, antitumoral, antimitotic, cytotoxic, and anti-infection activities, as reviewed in [5,12]. Due to their relatively low redox potential, they have a great probability of undergoing electron transfer reactions.

In our continuing search for new antibacterial phenolic compounds, we have synthesized and evaluated hundreds of compounds using antibacterial phenotypic assays against gram-positive and gram-negative species [13,14,15,16]. The structural design of our library of curcumin analogs has been based on phenolic functionality, because most of bacterial division inhibitors display at least one phenol ring [17,18]. Among the tested phenolic compounds, the most potent compounds were BC1 and T9A, which were tested against human pathogens and found to be active against gram-positive and gram-negative species (to be published elsewhere). The most active compound against *S. aureus*, T9A (Figure 1, hydroxychalcone 5, Appendix A), inhibited growth of both bacteria albeit with a 16-fold higher MIC_50_ for the gram-negative bacterium. Plating of treated cells revealed that the cells were still alive at all concentrations of T9A tested, thus a minimal bactericidal concentration (MBC) for this compound could not be determined, and T9A seems bacteriostatic but not bactericidal for *S. aureus*. As both BC1 and T9A display antimicrobial activity against the gram-negative phytopathogen *X. citri* (see below), we decided to further characterize their modes of action. Here, we tested the influence of T9A and BC1 on macromolecular synthesis (DNA, RNA, protein, and peptidoglycan), ATP concentration, and membrane permeability using *X. citri* and *B. subtilis*. We chose *B. subtilis* as a gram-positive model organism, as this is commonly used as a model for mode of action studies, and several methods are established in our labs, including FtsZ purification and FT-IR spectrophotometry [14].

## 2. Results

### 2.1. Chalcones Inhibit Growth of X. citri and B. subtilis

The minimum inhibitory concentrations (MIC) of BC1 and T9A for *X. citri* and *B. subtilis* were determined using the broth microdilution method (Table 1). Overall, *X. citri* (MICs of BC1 = 90 µg/mL, and T9A = 50 µg/mL) was slightly more resistant to the compounds when compared to *B. subtilis* (BC1 = 50 µg/mL, and T9A = 40 µg/mL). Nevertheless, both compounds inhibited *X. citri* and *B. subtilis* with values very close and within the same order of magnitude as the positive control kanamycin (~20 µg/mL). Moreover, both BC1 and T9A were bactericidal for *X. citri*, and only bacteriostatic for *B. subtilis* at their respective MICs. Compared to our earlier results (Appendix A), we note that *B. subtilis* seemed less resistant to T9A compared to the gram-positive model *S. aureus*, whereas *X. citri* was nearly as resistant to this compound as the gram-negative bacterium *P. aeruginosa.* Though the MIC_50_ values in Appendix A were assessed by determining the metabolic activity after incubation for 24 h using the Resazurin Microtiter Assay (REMA), the results cannot be directly compared, since short incubations already strongly affect the metabolic activity of the cells (see below). The MIC values determined here were used throughout to guide our investigations about the modes of action of the compounds.

### 2.2. BC1 Permeabilizes the Membrane of B. subtilis

The ability to disrupt the bacterial membrane was assessed by measuring PI permeability in the presence of the compounds. T9A had no effect on the membrane of both *B. subtilis* and *X. citri*, whereas BC1 did not affect *X. citri* but did permeabilize *B. subtilis* membranes (Table 2; Appendix A). The difference in permeabilization combined with the fact that BC1 has a slightly lower MIC for *B. subtilis* than for *X. citri*, suggests that BC1 mode of action may be different between the two organisms. It has to be noted that measured membrane permeabilization does not necessarily mean permanent damage to the membranes, as MICs for membrane active compounds that we measure are generally tenfold higher than the concentration at which 50% of cells have membrane damage [19], and *B. subtilis* still displayed several other activities (respiration, synthesis of ATP and macromolecules, below) under the applied conditions.

### 2.3. Effects on Macromolecular Synthesis

To investigate whether BC1 or T9A can inhibit the synthesis of macromolecules such as DNA, RNA, proteins, and peptidoglycan, we followed the incorporation of radiolabeled precursors of each of these macromolecules (Figure 2). Both compounds inhibited most pathways to some extent in both *X. citri* (Figure 2A) and *B. subtilis* (Figure 2B), although T9A did not affect DNA synthesis, and also had little to no effect on protein synthesis in *X. citri.* In all cases, BC1 is more effective at blocking macromolecular synthesis than T9A. Initially, we established MICs for all control antibiotics used in *B. subtilis* and *X. citri*, and used MIC as this was the concentration at which BC1 and T9A were tested [20]. To our surprise, all control antibiotics were quite effective in blocking their respective pathways in *B. subtilis*, but not in *X. citri*, where only tetracycline blocked protein synthesis almost completely. Increasing the concentration of control antibiotics to four times the MIC did not result in a stronger inhibition of incorporation of precursors [20]. As the inhibition observed in *X. citri* was often similar to the inhibition observed with the control, it is difficult to comment on the inhibition of pathways in *X. citri*. However, as in both bacteria various pathways were more or less inhibited by both compounds, it is obvious that these compounds do not target one specific macromolecular synthesis pathway.

### 2.4. Effects on Metabolic Activity and ATP Levels

To determine whether the inhibition of macromolecular synthesis was specific, or caused by overall retardation of growth, or cell death, the metabolic activity of the cells in the presence of BC1, T9A, and pathway specific antibiotics was measured using the REMA assay. This assay is based on the reduction of resazurin to resorufin as a result of bacterial respiration. *X. citri* or *B. subtilis* cells were exposed to the compounds/antibiotics and their metabolic activity was measured in 20 min intervals for up to 120 min (Figure 3, Appendix A). For *X. citri*, even though there was some inhibition of metabolic activity, this did not go below 75% of uninhibited activity, nor displayed a large decline over the 2 h period in which activity was followed. Thus, inhibition of synthesis pathways to levels below 70% in Figure 3 is not only caused by the inhibition of growth. For *B. subtilis*, the results were different, and rather puzzling. The compound T9A almost completely blocked metabolic activity, which is striking, as the effect of T9A on macromolecular synthesis in *B. subtilis* was not very pronounced (Figure 2a). BC1 reduced the respiration to about 60%.

The relative intracellular ATP levels were determined in *X. citri* and *B. subtilis* after 80 or 30 min treatment with the compounds, respectively (Figure 4). At 1x MIC, T9A and BC1 cause a decrease in *X. citri* ATP levels compared to the control, to a similar extent as their effect on overall respiratory activity. BC1, at 2x the MIC led to an almost complete loss of ATP, when compared to the control CCCP (Figure 4B). Strikingly, in *B. subtilis*, the ATP levels increased for BC1 and T9A at 1x MIC. However, T9A at 2x MIC led to an almost complete loss of ATP, which is in line with the complete block of respiration observed (Figure 4A). This suggests that, in the presence of either compound, *B. subtilis* cells may first switch to anaerobic metabolism, together with an inhibition of ATP-consuming pathways, after which, at higher concentrations of T9A, metabolic activity stops completely.

### 2.5. FT-IR Spectrophotometry

FT-IR of *B. subtilis* cells treated with BC1 and T9A was measured, together with a negative control and a positive control using nisin (Figure 5). FT-IR is a technique that can provide a fingerprint of different components of the bacterial cell, and thus can also report on the disruption of cellular structures [14]. The peaks shown in the spectrum represent perturbations of diverse chemical structures from compounds present in *B. subtilis* [21]. The peaks at 1083 cm^−1^ are due to stretching of C-O bonds present in glycogen. The peaks at 1231 cm^−1^ are due to PO_2_^−^ asymmetric stretching, mainly from nucleic acids with little contribution from phospholipids. The peaks at 1404 cm^−1^ corresponds to COO^−^ symmetric stretching from amino acid side chains and fatty acids. The peaks at 1450 cm^−1^ are due to CH_2_ bending from lipids present in the cytoplasmic membrane. The peaks at 1540 cm^−1^ are from N-H bending and C-N stretching from amide II present in α helixes of proteins. Finally, the peaks at 1646 cm^−1^ are from C=O stretching from amide I α helixes. The peaks of the spectrum of the negative control (Figure 5A) were similar to the spectrum of the cells treated with T9A (Figure 5D), and the peaks of the spectrum of cells treated with nisin (Figure 5B) were similar to peaks in the spectrum of the cells treated with BC1 (Figure 5C). We could not identify any obvious differences between the spectra regarding a particular peak. However, the similar intensities between the spectrum of the negative control and T9A and between BC1 and nisin, corroborates with the results of our membrane permeability assay (Table 2). There we can see that T9A does not cause membrane disruption in *B. subtilis*, whereas BC1 is able to disrupt the cellular membrane similar to nisin. We attempted to apply the FT-IR technique on *X. citri* cells as well. However, *X. citri* produces xanthan gum and biofilms [22], generating noise in the spectra, which prevents the detection of the peaks that report on cellular structures.

### 2.6. Cell Division Investigations

To assess the ability of the compounds to disrupt cell division, reporter strains expressing a functional fluorescent fusion to a cell division protein (GFP-ZapA in *X. citri* and FtsZ-GFP in *B. subtilis*) were treated with T9A and BC1, and imaged by microscopy (Figure 6). *X. citri* cells treated only with the vehicle (1% DMSO) showed a perpendicular fluorescent signal at mid-cell, which corresponds to the division septum (Figure 6A). Exposure of *X. citri* to both compounds had no apparent effect on cell division since we could still detect localized GFP-ZapA in the cells (Figure 6B,C). Finally, exposure to hexyl-gallate, our positive control [23], led to a complete delocalization of GFP-ZapA in *X. citri* (Figure 6D). Conversely with what was observed for *X. citri*, the localization of FtsZ-GFP, our marker for cell division in *B. subtilis* (Figure 6E), was affected by both BC1 and T9A (Figure 6F,G, respectively). Here, the fluorescence of FtsZ-GFP looks dispersed and cytosolic, similar to what can be seen for the control hexyl-gallate (compare 6F and 6G with 6H). Since BC1 is able to disrupt membrane permeability in *B. subtilis* (disrupting membrane potential as well), it is possible that FtsZ-GFP localization is disrupted due to membrane potential disruption [24]. However, T9A did not disrupt membrane permeability in *B. subtilis*, suggesting that the mode of action of T9A could be linked to cell division. Thus, we tested the effect of BC1 and T9A on the GTPase activity of purified *B. subtilis* FtsZ (Figure 7). As the GTPase activity of FtsZ was not significantly affected by BC1 or T9A in comparison with the negative control DMSO, it is unlikely that FtsZ is a direct target of the chalcones.

## 3. Discussion

*X. citri* causes Asiatic Citrus Canker, leading to economic losses to citriculture [1]. Here, we investigated the antibacterial mode of action of the methoxychalcone BC1 and the hydroxychalcone T9A on *X. citri* and *B. subtilis*, using techniques previously used to investigate the mode of action of drugs in model organisms such as *B. subtilis* itself, and also *E. coli* and *S. aureus* [25,26,27,28].

In *X. citri*, all the metabolic pathways are affected by the compounds within the 80 min monitoring, which is a strong indication of an aspecific mechanism. It has to be noted that most positive controls for pathway inhibition in *X. citri* were not very effective in blocking incorporation of metabolites—even though all the compounds used were confirmed as bona fide antibiotics [20]. Possibly, the slow growth of *X. citri* makes it difficult to see differences in incorporation; however, increasing control concentrations was not effective within the timeframe of the experiment. Longer exposure or higher concentrations of antibiotic would most likely lead to secondary effects, such as overall growth reduction, which makes it impossible to specify whether reduced incorporation is due to pathway inhibition. Inhibition of multiple pathways is similar to what was described by Haraguchi et al. [7] on the mode of action of licochalcone A. This chalcone inhibited incorporation of DNA, RNA, and protein (peptidoglycan was not tested) in *Micrococcus luteus*, via inhibition of oxygen consumption and interference with energy metabolism [7]. Similarly, for *B. subtilis* all the metabolic pathways are affected within the 60 min tested.

Intriguingly, in our experiments, we showed that both compounds (BC1 and T9A) caused an increase in ATP concentration in *B. subtilis* (Figure 4). This was surprising, as the observed membrane damage with BC1, and the drop in respiration for both compounds, would suggest a decrease in ATP production, which was only observed at higher concentrations of T9A, which collapsed ATP. It could be that both compounds cause a switch of *B. subtilis* to anaerobic growth, which allows ATP levels to be maintained [29], but the increase in ATP could also be due to processes such as cell wall and RNA synthesis being inhibited or blocked, causing a difference between ATP production and consumption. The latter effect was previously observed for the protein inhibitors chloramphenicol and streptomycin [30]. In *X. citri*, a decrease in ATP levels was observed (Figure 4), especially for BC1 at higher concentrations, which is an indication of indirect effect.

Cell division processes were investigated by observing the localization of GFP fusions with cell division proteins. In *X. citri* neither BC1 nor T9A caused delocalization of our cell division marker (GFP-ZapA) which rules out the possibility that these compounds cause a direct effect on *X. citri* cell division. In *B. subtilis* both compounds caused delocalization of our cell division marker (FtsZ-GFP). BC1 was able to permeabilize *B. subtilis* membranes, therefore this result could be due to membrane potential disruption, similar to that observed by Strahl, et al. [24]. However, T9A was not able to permeabilize *B. subtilis* membranes. Therefore, we tested FtsZ GTPase activity, and since neither chalcone caused an effect on GTPase activity, we rule out the possibility of FtsZ being directly affected by T9A or BC1.

Summarizing, the chalcones affect both gram-negative and gram-positive cells but not in the same manner. In gram-positive cells, the cytoplasmic membrane is a target; in gram-negative cells the outer membrane may be a sink for these compounds that would prevent them from reaching the cytoplasmic membrane. It remains to be established however, how this would result in the overall decrease in ATP levels and inhibition of macromolecular synthesis observed upon exposure of a gram-negative bacterium to these compounds.

## 4. Materials and Methods

### 4.1. Chemical Synthesis

Synthesis of BC1 was achieved by the reaction between acetovanillone (5 mmol) and benzaldehyde (5 mmol) at room temperature using ethanol as the solvent and ethanolic solution of NaOH (1.0 mol/L, 5 mL), using a protocol reported by Santos and collaborators with minor modifications [31]. The reaction mixture was stirred and monitored by successive thin-layer chromatography (TLC) analyses. When the reaction was finished, the residue was poured into crushed ice. The resulting precipitate was removed by filtration, and product obtained was purified by recrystallization from ethanol at 76 °C, with yield of 89%.

In order to synthesize T9A, acetophenone (5 mmol) and 2-hydroxybenzaldehyde (5 mmol) were solubilized in a NaOH ethanolic solution (1.0 mol/L, 20 mL) at room temperature, using protocols reported by Kobelnik and collaborators with minor modifications [32]. The reaction was monitored by TLC analyses. When reagents were converted to the product, the reaction medium was poured into crushed ice and acidified with HCl solution (1.0 mol/L) at pH 2. The precipitate was filtered and the product obtained was recrystallized in ethanol, with yield of 54%. All reagents were purchased from Merck (Kenilworth, NJ, USA).

### 4.2. Minimal Inhibitory Concentration Assays

The minimal inhibitory concentration (MIC) of the compounds against *X. citri* and *B. subtilis* was determined by broth microdilution. Exponentially growing bacteria were diluted to OD_600_ of 0.005 in 96 well plates containing twofold serial dilutions of the compounds, in a final volume of 200 µL of growth medium. Davis Minimal Medium (DMM) (casaminoacids 2.0 g/L; K_2_HPO_4_ 7.0 g/L; KH_2_PO_4_ 3.0 g/L; MgSO_4_·7(H_2_O) 0.1 g/L; (NH_4_)_2_SO_4_ 1.0 g/L; Na_3_C_6_H_5_O_7_·2(H_2_O) 0.5 g/L; glucose 7.0 g/L and tryptophan 10 mg/L), modified from [33,34], was used for *B. subtilis*. Xam1 medium (glycerol 2.46 g/L; MgSO_4_·7H_2_O 0.247 g/L; (NH_4_)_2_SO_4_ 1.0 g/L; K_2_HPO_4_ 10.5 g/L; KH_2_PO_4_ 4.5 g/L; Na_3_C_6_H_5_O_7_·2H_2_O 0.5 g/L; casaminoacids 0.3 g/L; BSA 1 g/L (pH 5.4 adjusted with HCl)) was used for *X. citri*. The MIC was defined as the lowest concentration of compound that prevented bacterial growth after incubating cultures for 24 h at 30 °C [35]. MICs were also determined using the Resazurin Microtiter Assay Plate (REMA) essentially as described by [14]. The antibiotic kanamycin at 20 µg/mL was used as positive control in both evaluations. Three independent experiments were used to estimate the MIC data. Statistics were done using Graphpad Prism version 6 (Graphpad Software, San Diego, CA, USA). The bacteriostatic or bactericidal action of compounds was assessed by plating samples from REMA on solid-media following incubation for 24 h (*B. subtilis*) or 48 h (*X. citri*) in order to check for cell viability.

### 4.3. Membrane Permeability Assay

Membrane permeability was tested using the Live/Dead BacLight bacterial viability kit (Thermo-Scientific Scientific, Waltham, MA, USA). An overnight culture of *B. subtilis* grown in LB, or *X. citri* grown in NYGB [15], at 29 °C, was diluted to 10^6^ cells/mL. *B. subtilis* cells were incubated in the presence of BC1 (50 µg/mL), T9A (40 µg/mL), DMSO (1%) as a negative control, and nisin (5.0 μg/mL) as a positive control; *X. citri* cells were incubated in the presence of BC1 (90 µg/mL), T9A (50 µg/mL), DMSO (1%) as a negative control, and 20 min at 55 °C as a positive control. Treatments were carried out for 15 min and 30 min at 23 °C. Propidium iodide (5.0 mM) and SYTO 9 (835 µM) were added to each test. After incubation, cells were mounted on agarose pads for microscopy analysis and imaged using an Olympus BX-61 microscope with an OrcaFlash-2.8 camera using DAPI/FITC and Texas Red fluorescence filter cubes. Propidium iodide (shown in Texas Red fluorescence filter) is membrane impermeable, and thus only stains cells with disrupted membrane, whereas SYTO 9 (shown in DAPI/FITC fluorescence filter) is membrane permeable and stains all the cells. The Olympus cellSens platform was used for image analysis. Experiments were performed in triplicates.

### 4.4. Macromolecular Synthesis

A total of four macromolecular synthesis pathways were evaluated by monitoring the incorporation of radioactively labelled precursors. [5-^3^H] uridine, [methyl-^3^H] thymidine, L-[3,4,5-^3^H(*N*)] leucine and D-[6-^3^H(*N*)] glucosamine hydrochloride (all at 0.5 µCi/mL) were used to respectively monitor RNA, DNA, protein and peptidoglycan synthesis. During this experiment *B. subtilis* cells were grown to early exponential phase in DMM and *X. citri* cells were grown to early exponential phase in Xam1. 1 mM of non-labelled precursor was used for RNA and peptidoglycan and 10 µM was used for DNA and protein. For DNA incorporation, *B. subtilis* cells were grown in LB medium instead of DMM. Cells were incubated in the presence of labelled precursors for 80 min, with shaking at 30 °C. *B. subtilis* was tested in the presence of 40 µg/mL of BC1 or T9A, without compound (negative control), or with an antibiotic known to inhibit the synthesis pathway tested as a positive control (RNA: rifampicin 0.625 µg/mL, DNA: ciprofloxacin 0.625 μg/mL, protein: tetracycline 10 μg/mL, peptidoglycan: vancomycin 1.0 μg/mL). *X. citri* was tested in the presence of 30 µg/mL of BC1, 20 µg/mL of T9A, without compound (negative control), or with an antibiotic supposed to inhibit the synthesis pathway tested as positive control (RNA: rifampicin 0.625 µg/mL, DNA: ciprofloxacin 10.0 μg/mL, protein: tetracycline 10 μg/mL, peptidoglycan: penicillin G 800 μg/mL). For every precursor tested, at least two technical replicates were taken, from three biological replicates (minimum six datapoints). During incubation, aliquots were taken and precipitated with ice-cold 12% trichloroacetic acid for 35 min and then filtered through nitrocellulose membranes (pore size 0.45 μm). The filters were washed with ice cold 12% trichloroacetic acid, transferred to 2 mL scintillation fluid Ultima Gold MV (PerkinElmer, Waltham, MA, USA), and measured in a Tri-Carb 2000CA liquid scintillation analyzer (Packard Instruments, part of Thermo-Scientific Scientific, Waltham, MA, USA).

### 4.5. Resazurin Assay

The Resazurin assay (REMA) was performed in order to evaluate the impact of the compounds on the respiratory activity after a short period of time [23,36]. Cells of *B. subtilis* or *X. citri* were incubated in contact with resazurin in the same medium used for the MIC evaluations (DMM and Xam1 respectively) in 96-well plates. The compound BC1 was used at a final concentration of 40 µg/mL for *B. subtilis* and 30 µg/mL for *X. citri*; T9A was used at 40 µg/mL for *B. subtilis* and 20 µg/mL for *X. citri*. Rifampicin 0.625 µg/mL, ciprofloxacin 0.625 μg/mL, tetracycline 10 μg/mL, and vancomycin 1.0 μg/mL were used as controls for *B. subtilis*, and rifampicin 0.625 µg/mL, ciprofloxacin 10.0 μg/mL, tetracycline 10 μg/mL, and penicillin G 800 μg/mL were used as controls for *X. citri*, along with samples without added compound. Resazurin was added to a final concentration of 0.1 mg/mL. Fluorescence (excitation 530 nm, emission 560 nm, bandwidth 9 nm) was recorded every 20 min up to 120 min in a BioTek Synergy Mx 96-well plate reader (BioTek instruments, Winooski, VT, USA).

### 4.6. Intracellular ATP Concentration Assay

ATP levels were measured in *B. subtilis* and *X. citri* using the BacTiter-Glo™ Microbial Cell Viability Assay (Promega). CCCP (Carbonyl cyanide *m*-chlorophenyl hydrazine, 40 µg/mL) was used as a positive control for ATP depletion. Cells were incubated in a 96 well plate (30 °C, 1000 rpm), 30 min for *B. subtilis* (in DMM) and 80 min for *X. citri* (in Xam1) after which 100 µL of cell culture was added to 100 µL of BacTiter-Glo™ Reagent for 5 min in a white 96 well plate (26 °C, 1000 rpm). Luminescence was measured in a Tecan Infinite F200 Pro luminometer. The amount of light emitted is a measure for the intracellular ATP concentration.

### 4.7. FT-IR Spectrophotometry

Fourier-transform infrared (FT-IR) spectrophotometry was done on *B. subtilis* cells in a FT-IR spectrophotometer (Shimadzu, model 8300, Shimadzu, Kyoto, Japan), as described by [14]. Cells were cultivated in LB medium (29 °C) up to the concentration of 10^5^ cells/mL, then treated with BC1 (50 µg/mL), T9A (40 µg/mL), nisin (5 µg/mL), or a negative control without the addition of any compound for 30 min. Aliquots of 1.5 mL of cells were then centrifuged at 10.000 x g for 2 min, washed with water, and centrifuged two more times to remove traces of medium. After that, the samples were air-dried, and mixed with 150 mg of KBr. Next, samples were compressed at 40 kN for 5 min. Absorbance was measured from 400 cm^−1^ to 4000 cm^−1^, with 32 scans at a resolution of 4 cm^−1^. Data treatment and analyses were performed using the software Origin 8.00 (OriginLab, Northampton, MA, USA).

### 4.8. Cell Division Interference

The effect of chalcones on cell division was investigated using GFP fusions to cell division proteins, and by evaluating the GTPase activity of FtsZ from *B. subtilis*. *X. citri* labeled with GFP-ZapA (*X. citri amy: gfp-zapA* [37] was grown in NYGB and induced with Xylose (0.5%) for 90 min. Cells were treated with BC1 (90 µg/mL), T9A (50 µg/mL), DMSO (1%, as a negative control) or hexyl gallate (60 µg/mL, as a positive control) for 30 min and samples were prepared for visualization as described before [37]. *B. subtilis* labelled with FtsZ-GFP [14] was cultivated in the presence of 0.02 mM IPTG (isopropyl-β-d-1-thiogalactopyranoside) to induce the expression of the protein fusion from the *P_spac_* promoter. Cells were treated with BC1 (50 µg/mL), T9A (40 µg/mL), DMSO (1%) as a negative control, and hexyl gallate (60 µg/mL) as a positive control for 15 and 30 min.

FtsZ from *B. subtilis* was expressed and purified using the ammonium sulfate precipitation method as described by [38]. The FtsZ GTP hydrolysis rate was determined using the malachite green phosphate assay as described by [38,39], with a few modifications. Stocks of BC1 (50 µg/mL), T9A (40 µg/mL), and DMSO (1%) as a negative control, with FtsZ (24 µM), MgCl_2_ (20 mM) and Triton X-100 (0.02%) prepared in polymerization buffer (HEPES 50 mM, KCl 300 mM, pH 7.5) were stabilized at 30 °C for 5 min. Then, GTP dissolved in the same buffer was added (1 mM final concentration) at different time points (0, 2.5, 5, 10, 15, 20, and 30 min). The reactions were kept at 30 °C and developed in accordance with Malachite Green Phosphate Assay Kit (MAK307, Sigma-Aldrich, St. Louis, MO, USA).

## 5. Conclusions

The chalcones described in this study affect both gram-negative and gram-positive cells but not in the same manner. In gram-positive cells, the cytoplasmic membrane is a target; in gram-negative cells the ATP levels drop and macromolecular synthesis is inhibited.

## Figures and Tables

**Figure 1 molecules-25-04596-f001:**
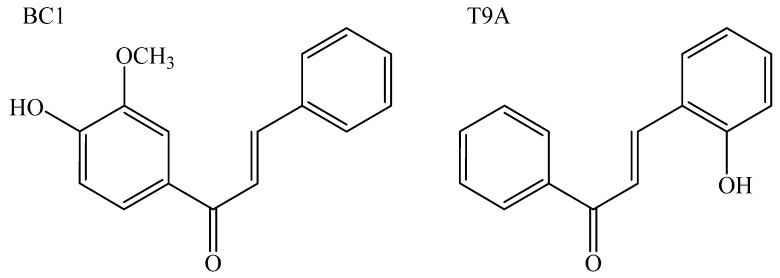
Structures of the compounds used in this study.

**Figure 2 molecules-25-04596-f002:**
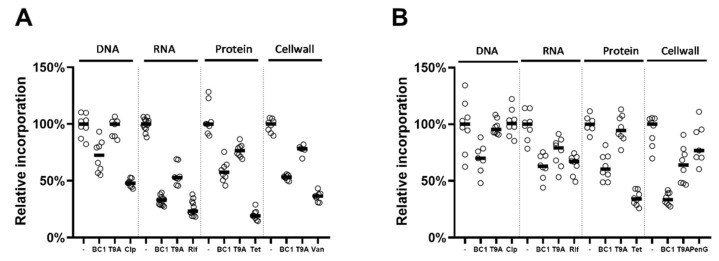
Effects of BC1 and T9A on the four major macromolecular synthesis pathways of *B. subtilis* (**A**), and *X. citri* (**B**), in comparison to negative controls (−) and antibiotics with specific activities targeting DNA, RNA, Protein, and Peptidoglycan syntheses, Ciprofloxacin (Cip), Rifampicin (Rif), Tetracycline (Tet), and (A) Benzyl penicillin (PenG), (B) Vancomycin (Van), respectively. Each circle represents an experimental replicate; bars indicate the median relative incorporation at each condition tested.

**Figure 3 molecules-25-04596-f003:**
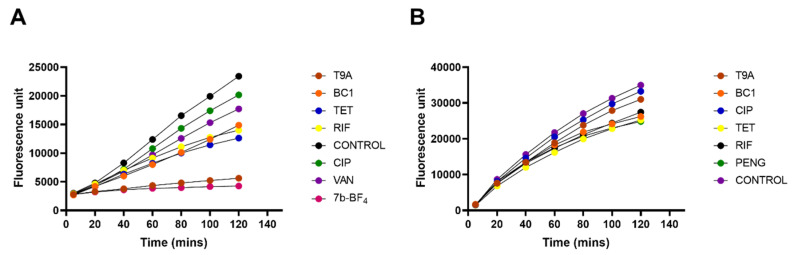
Assessment of the overall metabolic activity of *B. subtilis* and *X. citri* exposed to BC1 and T9A. Absolute fluorescence of resorufin measured over time in 20 min intervals in *B. subtilis* (**A**), and *X. citri* (**B**) cells. Fluorescence relative to the control is depicted in Appendix A. Control compound 7b-BF_4_ is a compound that completely blocks *B. subtilis* respiration [20].

**Figure 4 molecules-25-04596-f004:**
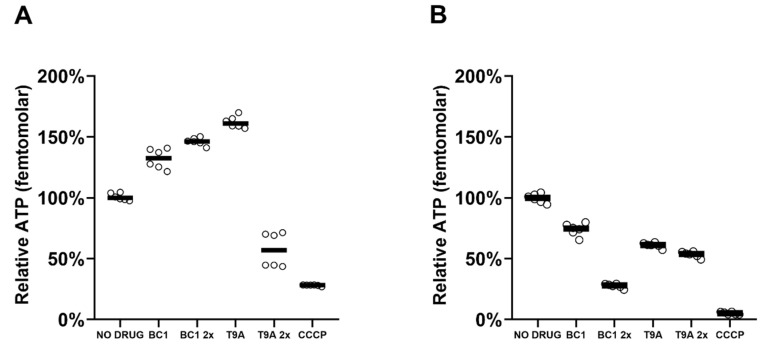
ATP concentration (relative light units): (**A**) after 30 min treatment of *B. subtilis* cells and (**B**) after 80 min treatment of *X. citri* cells, with 1x and 2x the MIC of BC1 or T9A or 40 µg/mL of CCCP. 100% is defined as the median luminescence of the control sample (without any compound added); each circle represents a replicate, and bars indicate the median of each condition tested.

**Figure 5 molecules-25-04596-f005:**
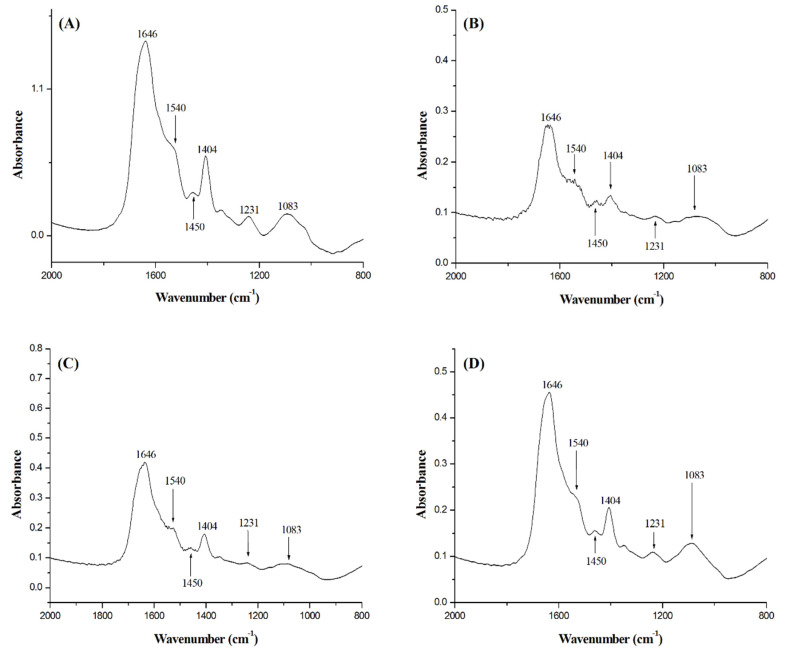
FT-IR spectra of *B. subtilis* cells after 30 min treatment with (**A**) no compounds added; (**B**) nisin (5 µg/mL); (**C**) BC1 (50 µg/mL); and (**D**) T9A (40 µg/mL).

**Figure 6 molecules-25-04596-f006:**
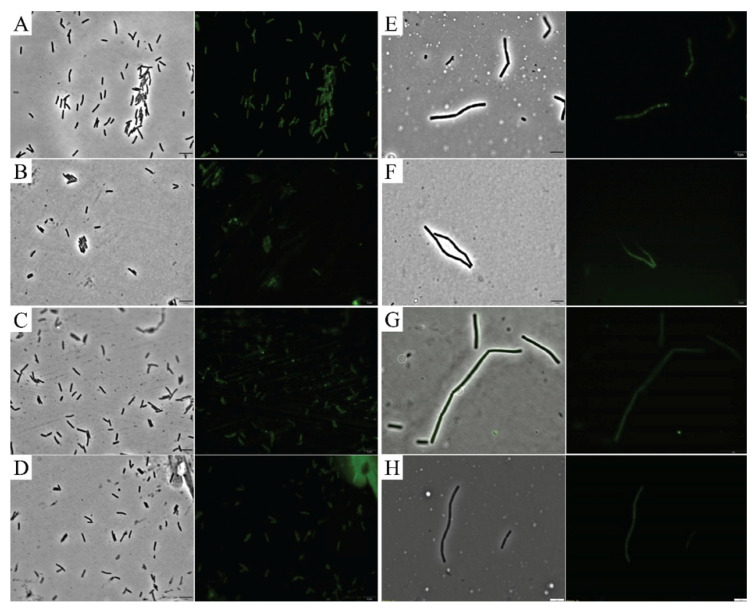
Effects of BC1 and T9A on bacterial cell division. Phase contrast (left) and GFP (right) images of: *X. citri* expressing GFP-ZapA treated for 30 min with (**A**) 1% DMSO, (**B**) BC1 (90 µg/mL), (**C**) T9A (50 µg/mL), (**D**) hexyl gallate (60 µg/mL), and *B. subtilis* expressing FtsZ-GFP treated for 30 min with (**E**) 1% DMSO, (**F**) BC1 (50 µg/mL), (**G**) T9A (40 µg/mL), (**H**) hexyl gallate (60 µg/mL). Scale bar: 5 µm.

**Figure 7 molecules-25-04596-f007:**
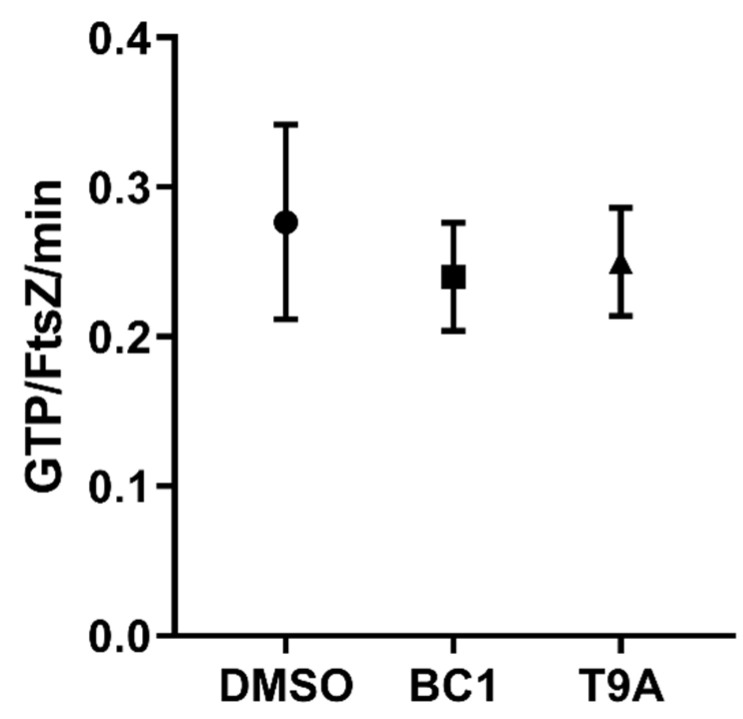
GTP hydrolysis activity of *B. subtilis* FtsZ in the presence of DMSO, BC1, or T9A.

**Table 1 molecules-25-04596-t001:** Minimal inhibitory concentrations of the compounds (µg/mL).

Compound	*X. citri*	*B. subtilis*
BC1	90 *	50 ^#^
T9A	50 *	40 ^#^

* bactericidal; ^#^ bacteriostatic.

**Table 2 molecules-25-04596-t002:** Membrane permeabilization of *X. citri* and B. subtilis cells by chalcones.

Treatment	Permeabilized *X. citri* Cells (Mean ± SD)	Permeabilized *B. subtilis* Cells (Mean ± SD)
Negative control	2.38 ± 0.08%	2.86 ± 1.78%
DMSO 1%	2.95 ± 0.53%	4.02 ± 3.16%
BC1 (15 min)	2.22 ± 1.34%	66.23 ± 0.62%
BC1 (30 min)	6.98 ± 1.67%	75.32 ± 0.28%
T9A (15 min)	2.91 ± 0.89%	3.67 ± 0.63%
T9A (30 min)	3.86 ± 0.22%	4.81 ± 3.29%
Heat shock (60 °C; 20 min)	97.16 ± 0.76%	not done
Nisin (5.0 µg/mL; 15 min)	not done	97.22 ± 2.72%

Results from two independent replicates with 200 cells each. Concentrations used: BC1 90 µg/mL (*X. citri*) or 50 µg/mL (*B. subtilis*); T9A 50 µg/mL (*X. citri*) or 40 µg/mL (*B. subtilis*).

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
