# Peer review of "Investigating the Modes of Action of the Antimicrobial Chalcones BC1 and T9A"

_molecules, 2020, doi:10.3390/molecules25204596_

Round 1

Reviewer 1 Report

In this paper the Authors attempted to describe the mechanism of antibacterial action of two  selected chalcones compounds (BC1 and T9A) against Xanthomonas citri, a phytopathogen which causes Asiatic Citrus Canker, and Bacillus subtilis, as a model of Gram-positive bacteria.The Authors studied the effects of these compounds on the synthesis of macromolecules, on the cytoplasmic membrane, on metabolic activity and cell division. The Authors observed that two compunds possess different mechanism of action. In particular, T9A had no effect on the membrane of both B. subtilis and X. citri whereas BC1 did not affect X. citri but did permeabilize B. subtilis membranes. Both compounds inhibited most pathways to some extent in both X. citri and B. subtilis. In all cases BC1 is more effective at blocking macromolecular synthesis than T9A. Both compounds had no effect on cell division of X. citri whereas B. subtilis was affected by both BC1 and T9A.

The conclusions of Authors are that in Gram-positive cells the cytoplasmic membraneis a target of these compounds whereas in Gram-negative cells the outer membrane would prevent the compounds to reach the cytoplasmic membrane. It remains to be established how the chalcones cause a decrease in ATP levels and inhibition of macromolecular synthesis observed upon exposure of  Gram-negative bacteria.

Comments:

The topic of the paper is interesting and directed to a specific application. Undoubtedly, the identification of new antibacterial compounds is needed both in agriculture field and for human application.  

The paper is well written and the methodology used is adequate with respect to the study and the conclusions obtained.

The Authors clarified some aspects of the mechanism of action of the chalcones considered, although further studies will be needed to completely clarify the exact  mechanism of action of these compounds.

Minor comments:

  • In the introduction, specify that X. citri is a Gram-negative bacterium
  • In the paragraph 2.1 (results)  lines 87-89, the sentence is wrong.
  • " sensitive" should be substituted by "resistant"
  • In figure 3 the colours of circles are not distinguishable. Change some colours and enlarge the circles.

Author Response

We would like to thank the reviewer for their comments. We have made the following changes based on the comments (answers in italics).

Minor comments:

  • In the introduction, specify that X. citri is a Gram-negative bacterium – Done, see addition in line 80.
  • In the paragraph 2.1 (results)  lines 87-89, the sentence is wrong. – Correct, we should have removed the sentence about REMA and polynomial regression as this was done for the other bacteria in Table S1. The confusing sentence has been removed.
  • " sensitive" should be substituted by "resistant" – We have made the replacements.
  • In figure 3 the colours of circles are not distinguishable. Change some colours and enlarge the circles. –  Fig. 3 has been replaced by a figure with larger circles.

Reviewer 2 Report

The chalcone topic is back in fashion and the authors have strongly contributed to elucidate some open questions, that unfortunately remain outstanding. In the last year, (2019-2020) several papers regarding chalcones and their derivatives have been published without providing any novelty in this topic. Moreover the template is always the same: take one or two chalcone derivatives and test on bacteria. Despite the experimental part shown in the paper is detailed and focused on the aim, I'm curious about the novelty that the authors want to provide in this topic, it is not so evidenced. Despite this, I have some questions:

1) line 84: please explain the abbreviation

2) line 72: why the MBC of compound T9A couldn't determine against S. aureus? It is not so clear for me

3) The authors used two bacteria: X. citrus (Gram -) the main target of the paper and B. subtilis (Gram +), without motivating the chosen of this latter and the reason for the comparison.

4) It's well noted that the bacteria gram+ and - present different structural   and physiological characteristics, starting from the membrane, which have been oriented medicinal chemists and chemists to synthesize compounds selective for each bacterium (+) and (-). Why the authors have decided to test only two molecules with structural similarities? without: I) any comment on the chosen of the synthesis of these two compounds and on ii) the Structure active relationship (SAR). Can the author think that there is any correlation between the chalcone structure and the various target that they have tested for both bacteria? Can have any influences on the action mechanism?

I attached a review that can be helpful for motivate the response. Chem Rev. 2017 Jun 28; 117(12): 7762–7810.

4) As the authors have highlighted, the result on the X.citrus seems to be discouraging although significant. In the discussion section, I noted a supported discussion regarding the result obtained for s. subtilis and a meager one for x.citrus. I suggest the authors to divide the discussion for the two bacteria given that any correlation between them exists. 

Author Response

We would like to thank the reviewer for their comments. We have made the following changes based on the comments (answers in italics).

1) line 84: please explain the abbreviation – Both the abbreviation MIC and REMA are now written out in this paragraph.

2) line 72: why the MBC of compound T9A couldn't determine against S. aureus? It is not so clear for me. – S. aureus cells exposed to T9A for 24 h did not display growth, but when subsequently plated still started growing – thus the compound is bacteriostatic, not -bactericidal. The sentence has been rephrased to make this more clear.

3) The authors used two bacteria: X. citrus (Gram -) the main target of the paper and B. subtilis (Gram +), without motivating the chosen of this latter and the reason for the comparison. – We have included a sentence to explain this better (line 83): “We chose B. subtilis as a Gram-positive model organism as this is commonly used as a model for mode of action studies and several methods are established in our labs including FtsZ purification and FT-IR spectrophotometry [14].”

4) It's well noted that the bacteria gram+ and - present different structural   and physiological characteristics, starting from the membrane, which have been oriented medicinal chemists and chemists to synthesize compounds selective for each bacterium (+) and (-). Why the authors have decided to test only two molecules with structural similarities? without: I) any comment on the chosen of the synthesis of these two compounds and on ii) the Structure active relationship (SAR). Can the author think that there is any correlation between the chalcone structure and the various target that they have tested for both bacteria? Can have any influences on the action mechanism? I attached a review that can be helpful for motivate the response. Chem Rev. 2017 Jun 28; 117(12): 7762–7810.

We have rephrased part of the introduction (line 68-74) to better explain how the synthesis of these compounds fits with previous compounds synthesized and tested, and why we aim for compounds with at least a phenolic ring.

4) As the authors have highlighted, the result on the X.citrus seems to be discouraging although significant. In the discussion section, I noted a supported discussion regarding the result obtained for s. subtilis and a meager one for x.citrus. I suggest the authors to divide the discussion for the two bacteria given that any correlation between them exists. – We appreciate the comment but also tried to keep the discussion short, which is best done by discussing the experiments in order and highlighting the most important results per mode-of-action experiment. We slightly changed the order of the discussion of the biosynthetic pathways to make the separation between both organisms more clear.

Round 2

Reviewer 2 Report

The revisions made by the authors matched well with the required indications. The additional information makes the text quite understandable.